# Pursuer's Control Strategy for Orbital Pursuit-Evasion-Defense Game with Continuous Low Thrust Propulsion

**Junfeng Zhou, Lin Zhao, Jianhua Cheng \*, Shuo Wang and Yipeng Wang**

College of Automation, Harbin Engineering University, Harbin 150001, China
\* Correspondence: chengjianhua@hrbeu.edu.cn; Tel.: +86-156-4512-3923

**Abstract:** This paper studies the orbital pursuit-evasion-defense problem with the continuous low thrust propulsion. A control strategy for the pursuer is proposed based on the fuzzy comprehensive evaluation and the differential game. First, the system is described by the Lawden's equations, and simplified by introducing the relative state variables and the zero effort miss (ZEM) variables. Then, the objective function of the pursuer is designed based on the fuzzy comprehensive evaluation, and the analytical necessary conditions for the optimal control strategy are presented. Finally, a hybrid method combining the multi-objective genetic algorithm and the multiple shooting method is proposed to obtain the solution of the orbital pursuit-evasion-defense problem. The simulation results show that the proposed control strategy can handle the orbital pursuit-evasion-defense problem effectively.

**Keywords:** differential game; fuzzy comprehensive evaluation; continuous low thrust; zero effort miss variables

## 1. Introduction

Recently, the orbital pursuit-evasion problem has attracted increasing attention in space research [1–4]. This problem can be formulated as a differential game [5], which aims to obtain the optimal control strategy of the pursuer and/or the evader in the worst-case scenario, so as to realize the interception of the evader or the evasion from the pursuer.

Wong [6] was regarded as the first person to study the orbital pursuit-evasion problem, he solved the problem of intercepting a maneuverable satellite under the assumption of planar motion and constant gravitational field. Since then, many works have focused on the orbital pursuit-evasion problem. In reference [7], a method based on periodically updating the solution of the two-point boundary value problem (TPBVP) was proposed to generate near optimal feedback controls for the orbital pursuit-evasion problem. However, this method is time-consuming and difficult to be applied in real time. In order to overcome these drawbacks, Anderson [8] used a modified first-order differential dynamic programming algorithm to generate near-optimal feedback controls. References [9–11] found the saddle-point equilibrium solutions of the three-dimensional orbital pursuit-evasion game respectively by three different hybrid numerical methods. Hafer et al. [12] applied the sensitivity method to the orbital pursuit-evasion problem, which greatly reduces the computation burden for solving this problem numerically. Widhalm studied the problem of avoiding an interception and proposed two optimal evasive-maneuver strategies with the impulsive thrust [13] and the continuous low thrust [1] respectively. Prussing et al. [14] derived minimum-fuel impulsive strategies for return-on-state maneuvers by applying the primer vector theory. Merz [15] developed the guidance laws for the noisy satellite pursuit-evasion game. Woodbury et al. [16] studied an incomplete, imperfect

information game and presented the adaptive strategies for the pursuer and the evader. Ghosh et al. [17] developed a near-optimal feedback controller for the two-player pursuit-evasion games by using a new extremal-field approach. The above works were studied in the two-player pursuit-evasion game framework. However, in this framework, the evader can only perform maneuvers by itself to avoid threats. It is called self-defense, which disturbs the original mission of the evader and requires a large additional amount of fuel.

To overcome this disadvantage, a defender is introduced in [18]. The role of the defender is intercepting the pursuer. In this way, the evader can perform its original mission without being disturbed. A hybrid method combined particle swarm optimization with a Newton-Interpolation algorithm was proposed to solve the orbital defense problem. However, because of the introduction of the defender, the pursuer must avoid the interception by the defender while capturing the evader [19], which makes the design of the pursuer's control strategy more complicated. In order to develop control strategies for pursuers, Liu et al. [19] proposed a distributed online mission plan algorithm for pursuers to access targets. However, these works on the orbital pursuit-evasion-defense game adopted the impulsive thrust, which suffers the drawback that the interception will fail when the target can perform evasive maneuvers [4].

Compared with the impulse thrust, the continuous low thrust allows players to perform multiple, continuous maneuvers, which meets the requirements of the frequently orbital transfers in the game. When applying the continuous low thrust, the hypothesis about players' maneuverable is removed. It is closer to the actual situation of the orbital pursuit-evasion-defense game. Therefore, in this paper, the orbital pursuit-evasion-defense game model is constructed based on the continuous low thrust. Different from the model based on impulse thrust, the model based on continuous low thrust cannot adopt the Keplerian dynamics [20]. Its dynamic equations are based on the non-Keplerian motion. Two issues need to be solved in this model: (i) The system has a high dimension, which means that it will suffer from the curse of dimensionality [21] when solving the problem; (ii) two objectives, intercepting the evader and evading the defender, should be considered by the pursuer, and the corresponding weights should be determined according to the current state. For the first issue, as the zero effort miss (ZEM) can be used to simplify the linear system [4], the dimension of the system is reduced by introducing the relative state variables and the ZEM variables [22]. For the second issue, the pursuer's objective function is designed based on the fuzzy comprehensive evaluation, and the pursuer's control strategy which is suitable for the orbital pursuit-evasion-defense game is proposed. Based on the above model, the orbital pursuit-evasion-defense game is transformed into a TPBVP by applying the differential game theory. A hybrid method combining the multi-objective genetic algorithm and the multiple shooting method is presented to solve the TPBVP.

## 2. Mathematical Model of Orbital Pursuit-Evasion-Defense Game

### 2.1. Relative Orbital Dynamics

The orbital pursuit-evasion-defense game occurs in the final phase of the confrontation when the spacecraft are close enough so that they can identify each other with onboard electronic devices [4]. In this type of situation, the motion between the spacecraft can be expressed as relative motion [23]. As is known, Lawden's equations [24] and Clohessy–Wiltshire (C–W) equations [25] are two linearized equations used to describe the relative motion between spacecrafts. Unlike the C–W equations, which can only be applied to circular orbits, the Lawden's equations can describe the relative motion of a spacecraft in elliptical orbits. Same as in [26], the dynamics of the participating spacecraft are described in the local-vertical local-horizontal (LVLH) frame centered at a virtual spacecraft. In addition, Lawden's equations are adopted as the relative dynamic equations of the spacecraft.

As shown in Figure 1, $P, D, E$ respectively represents the pursuer, the defender, and the evader. We establish an elliptical fictitious spacecraft $O$ which is close to the players. The LVLH coordinate system is centered at the point $O$. $OX$ is pointing outward along the radius of the Earth, $OY$ is perpendicular

to *OX* in the reference orbital plane and points to the front of its flight direction, *OZ* is perpendicular to the orbital plane and forms a right-handed frame with *OX* and *OY*.

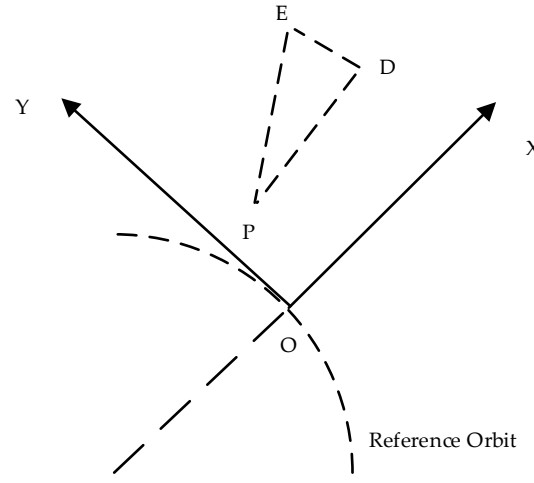

**Figure 1.** The local-vertical local-horizontal (LVLH) coordinate system.

The Lawden's equations can be expressed as:

$$\begin{cases} \ddot{x}_i = \omega^2 x_i + 2\omega\dot{y}_i + \dot{\omega}y_i + 2\frac{\mu x_i}{r_t^3} + T_i u_{xi} \\ \ddot{y}_i = -2\omega\dot{x}_i - \dot{\omega}x_i + \omega^2 y_i - \frac{\mu y_i}{r_t^3} + T_i u_{yi} \ , \qquad i = P, D, E \\ \ddot{z}_i = -\frac{\mu z_i}{r_t^3} + T_i u_{zi} \end{cases} \qquad (1)$$

where $\mu$ is the Earth gravitational constant, $r_t$ is the distance between the origin *O* and the Earth core, $\omega$ and $\dot{\omega}$ represent the orbital angular velocity and acceleration of the origin *O*, respectively. $x_i$, $y_i$, and $z_i$ represent the position components of the players in the relative coordinate system. $T_i$ represents the maximum thrust. $u_{xi}$, $u_{yi}$ and $u_{zi}$ respectively represent control variables in three directions (i.e., *x, y, z* axis), ranging from 0 to 1.

The state variables (i.e., position and velocity) of the players are represented by $X_i$ as follows:

$$X_i = [x_i, y_i, z_i, \dot{x}_i, \dot{y}_i, \dot{z}_i]^{\mathrm{T}}, \qquad i = P, D, E \qquad (2)$$

Thus, the dynamics equations can be written as:

$$\dot{X}_i = A_i X_i + T_i U_i \qquad (3)$$

where

$$A_i(t) = \begin{bmatrix} 0 & 0 & 0 & 1 & 0 & 0 \\ 0 & 0 & 0 & 0 & 1 & 0 \\ 0 & 0 & 0 & 0 & 0 & 1 \\ \omega^2 + \frac{2\mu}{r_t^3} & \dot{\omega} & 0 & 0 & 2\omega & 0 \\ -\dot{\omega} & \omega^2 - \frac{\mu}{r_t^3} & 0 & -2\omega & 0 & 0 \\ 0 & 0 & -\frac{\mu}{r_t^3} & 0 & 0 & 0 \end{bmatrix}, \quad i = P, D, E \qquad (4)$$

$U_i$ ($i = P, D, E$) is the control variable, which can be represented by

$$U_i = [0, 0, 0, u_{xi}, u_{yi}, u_{zi}]^{\mathrm{T}}, \quad \|U_i\| \le 1 \qquad (5)$$

## 2.2. Dimension-Reduction

According to Equation (3), each player has 6 state variables, so the total number of state variables in the game is 18. In the numerically solving process, co-state variables associated with state variables are introduced, and the problem converts to a 36-dimensional TPBVP. However, solving this high-dimensional TPBVP is quite difficult and possesses high computational demands [27]. In order to improve computational efficiency, the dimension of the system needs to be reduced. This process is performed in two steps. First, the relative state variables between the spacecraft are used to replace the system states. Then, the ZEM variables are applied to further reduce the number of variables and equations in the system.

In the first step, the game can be divided into two parts: One is the game between the pursuer $P$ and the evader $E$; the other is the game between the defender $D$ and the pursuer $P$. The relative state variables in two parts, $X_{PE}$ and $X_{DP}$, can be respectively represented by

$$\begin{cases} X_{PE} = X_P - X_E \\ X_{DP} = X_D - X_P \end{cases} \tag{6}$$

Substituting Equation (6) into Equation (3), the state equations are converted to:

$$\begin{cases} \dot{X}_{PE} = AX_{PE} + T_P U_P - T_E U_E \\ \dot{X}_{DP} = AX_{DP} + T_D U_D - T_P U_P \end{cases} \tag{7}$$

where $A = A_P = A_E = A_D$.

In the second step, according to the linear system theory, the zero-input state transfer matrix $\Phi(t_f, t)$ of the state equation is defined as:

$$\begin{cases} \Phi(t_f, t) = -\Phi(t_f, t)A \\ \Phi(t_f, t_f) = I_6 \end{cases} \tag{8}$$

where $t_f$ is the terminal time and $I_6$ is the $6 \times 6$ unit matrix.

Although two factors, the relative position and the relative velocity are involved in the game, only the first factor needs to be considered at the end of the game. The ZEM is the miss distance if both players do not apply any control from the current moment to the end of the game. Thus, the ZEM variables are introduced to reduce the dimension of the system and defined as:

$$\begin{cases} Z_{PE}(t) = D\Phi(t_f, t)X_{PE} \\ Z_{DP}(t) = D\Phi(t_f, t)X_{DP} \end{cases} \tag{9}$$

where $D = [I_{3\times3}, 0_{3\times3}]$.

Substituting Equation (9) into Equation (7), the state equations are reduced to:

$$\begin{cases} \dot{Z}_{PE} = D\Phi T_P U_P - D\Phi T_E U_E \\ \dot{Z}_{DP} = D\Phi T_D U_D - D\Phi T_P U_P \end{cases} \tag{10}$$

## 2.3. Design of Objective Function Based on Fuzzy Comprehensive Evaluation

In the orbital pursuit-evasion-defense game, the pursuer must survive from the defender's interception before it can successfully access the evader. Therefore, the pursuer-evader game and the defender-pursuer game must be considered and weighed in the objective function of the pursuer. As noted by Liu [19], the fuzzy comprehensive evaluation is an effective way to quantify various factors that are difficult to evaluate. Thus, it is used to obtain the weights corresponding to the two games. Detailed design is shown as follows.

By taking the terminal miss distance as the cost, the objective function of the three players can be defined as:

$$
\begin{cases}
J_E = -\|\mathbf{Z}_{PE}(t_f)\| \\
J_D = \|\mathbf{Z}_{DP}(t_f)\| \\
J_P = k_1\|\mathbf{Z}_{PE}(t_f)\| - k_2\|\mathbf{Z}_{DP}(t_f)\|
\end{cases}
\tag{11}
$$

where the parameter $k_i$, $i = 1, 2$ is the weight factor, which satisfies $k_i \geq 0$. $k_1 > k_2$ indicates that the pursuer prefers to reduce the terminal miss distance of the pursuer-evader game, while $k_1 < k_2$ indicates that the pursuer prefers to increase the terminal miss distance of the defender-pursuer game. The value of $k_i$ is divided into 11 scales, which are shown in Table 1.

**Table 1.** The evaluation scales.

| $v_i$ [1] | 1 | 2 | 3 | 4 | 5 | 6 | 7 | 8 | 9 | 10 | 11 |
|---|---|---|---|---|---|---|---|---|---|---|---|
| $k_1$ | 0 | 0.1 | 0.2 | 0.3 | 0.4 | 0.5 | 0.6 | 0.7 | 0.8 | 0.9 | 1 |
| $k_2$ | 1 | 0.9 | 0.8 | 0.7 | 0.6 | 0.5 | 0.4 | 0.3 | 0.2 | 0.1 | 0 |

[1] $v_i, i = 1, \cdots, 11$ represents the corresponding scales, respectively.

According to the analysis above, two factors need to be evaluated, one is the urgency of intercepting the evader at the very moment, denoted by $u_1$; the other is the urgency of evading the defender at the very moment, denoted by $u_2$. The effect of the factor $u_1$ increases as the ZEM distance of the pursuer-evader game decreases. The effect of the factor $u_2$ increases as the ZEM distance of the defender-pursuer game decreases. According to this rule, $\mathbf{Z}_{PE}(t)$ and $\mathbf{Z}_{DP}(t)$ are used to construct the weights of the two factors, which are given by:

$$
\begin{cases}
a_1 = 1 - \left(\dfrac{\|\mathbf{Z}_{PE}(t)\|}{\|\mathbf{Z}_{PE}(t)\| + \|\mathbf{Z}_{DP}(t)\|}\right)^3 \\
a_2 = 1 - a_1
\end{cases}
\tag{12}
$$

where $a_1$ and $a_2$ represent the weights of the factor $u_1$ and the factor $u_2$ respectively. Then the weight vector is expressed as: $\mathbf{A} = [a_1, a_2]$.

In order to establish the relationship between weighting factors and evaluation scales, the membership degree of each factor is calculated by the non-linear membership function which is written as follows:

$$
\begin{cases}
u_1(x) = (k(x-1))^3 \\
u_2(x) = 1 - (k(x-1))^3
\end{cases}
\tag{13}
$$

where $k = 0.1$, $x = 1, \cdots, 11$ are the corresponding evaluation scales.

Let $r_{ij} = u_i(j)$, where $i = 1, 2, j = 1, \cdots 11$, the fuzzy evaluation matrix can be obtained as:

$$
\mathbf{R} = [r_{ij}]_{2 \times 11}
\tag{14}
$$

The fuzzy comprehensive evaluation result vector is generated by the fuzzy synthetic operation of the weight vector and the fuzzy evaluation matrix. The fuzzy synthetic formula is defined as follows:

$$
\mathbf{B} = \mathbf{A} \circ \mathbf{R} = (b_1, b_2, \cdots, b_{11})
\tag{15}
$$

where "$\circ$" is a fuzzy synthetic operator. In this paper, the weighted average fuzzy arithmetic operator is adopted, which can make full use of the information of $\mathbf{R}$. It is specifically expressed as:

$$
b_j = \min\left\{1, \sum_{i=1}^{2} a_i \cdot r_{ij}\right\}, \quad j = 1, 2, \cdots, 11
\tag{16}
$$

The comprehensive evaluation value is obtained by analyzing the fuzzy comprehensive evaluation result vector. The analysis is done in the following steps. First, the result vector is normalized:

$$b'_j = \frac{b_j}{\sum\limits_{j=1}^{11} b_j} \tag{17}$$

Then, the normalized vector: $\boldsymbol{B'} = (b'_1, b'_2, \cdots, b'_n)$, which is processed using the weighted average principle. The evaluation value can be obtained as follows:

$$b = \frac{\sum\limits_{j=1}^{11} (b'_j)^k \cdot j}{\sum\limits_{j=1}^{11} (b'_j)^k} \tag{18}$$

where $k = 10$ is a specific coefficient. The purpose of this coefficient is to control the role played by a larger $b'_j$ $(j = 1, 2, \cdots, 11)$. If its value increases, the role of the largest term in $b'_j$ $(j = 1, 2, \cdots, 11)$ will be more prominent.

Finally, the values of $k_1$ and $k_2$ are obtained by finding the evaluation scale corresponding to the evaluation value $b$.

## 3. Solution Method for Orbital Pursuit-Evasion-Defense Game

### 3.1. Necessary Conditions for Optimal Strategies

The orbital pursuit-evasion-defense model given in the second section can be formulated as a non-cooperative N-person differential game. Necessary conditions for optimal strategies in this game are provided by Sarma [28] and applied to the system composed of (7) and (8) to obtain the form of optimal strategies.

The Hamiltonian function is introduced as follows:

$$\begin{cases} H_E = \boldsymbol{\lambda}_E^T \dot{\boldsymbol{Z}}_{PE} = \boldsymbol{\lambda}_E^T (\boldsymbol{D\Phi} T_P \boldsymbol{U}_P - \boldsymbol{D\Phi} T_E \boldsymbol{U}_E) \\ H_D = \boldsymbol{\lambda}_D^T \dot{\boldsymbol{Z}}_{DP} = \boldsymbol{\lambda}_D^T (\boldsymbol{D\Phi} T_D \boldsymbol{U}_D - \boldsymbol{D\Phi} T_P \boldsymbol{U}_P) \\ H_P = \boldsymbol{\lambda}_{PE}^T \dot{\boldsymbol{Z}}_{PE} + \boldsymbol{\lambda}_{DP}^T \dot{\boldsymbol{Z}}_{DP} = \boldsymbol{\lambda}_{PE}^T (\boldsymbol{D\Phi} T_P \boldsymbol{U}_P - \boldsymbol{D\Phi} T_E \boldsymbol{U}_E) + \boldsymbol{\lambda}_{DP}^T (\boldsymbol{D\Phi} T_D \boldsymbol{U}_D - \boldsymbol{D\Phi} T_P \boldsymbol{U}_P) \end{cases} \tag{19}$$

where $\boldsymbol{\lambda}_i$ $(i = P, D, PE, DP)$ are the co-state variables of the system.

According to the necessary conditions, the co-state equations are expressed as follows:

$$\begin{cases} \dot{\boldsymbol{\lambda}}_{PE} = -(\frac{\partial H_P}{\partial \boldsymbol{Z}_{PE}})^T = 0 \\ \dot{\boldsymbol{\lambda}}_{DP} = -(\frac{\partial H_P}{\partial \boldsymbol{Z}_{DP}})^T = 0 \\ \dot{\boldsymbol{\lambda}}_E = -(\frac{\partial H_E}{\partial \boldsymbol{Z}_{PE}})^T = 0 \\ \dot{\boldsymbol{\lambda}}_D = -(\frac{\partial H_D}{\partial \boldsymbol{Z}_{DP}})^T = 0 \end{cases} \tag{20}$$

and the transversality conditions are written as follows:

$$\begin{cases} \boldsymbol{\lambda}_{PE}(t_f) = \frac{\partial J_P}{\partial \boldsymbol{Z}_{PE}(t_f)} = k_1 \frac{\boldsymbol{Z}_{PE}(t_f)}{\|\boldsymbol{Z}_{PE}(t_f)\|} \\ \boldsymbol{\lambda}_{DP}(t_f) = \frac{\partial J_P}{\partial \boldsymbol{Z}_{DP}(t_f)} = -k_2 \frac{\boldsymbol{Z}_{DP}(t_f)}{\|\boldsymbol{Z}_{DP}(t_f)\|} \\ \boldsymbol{\lambda}_E(t_f) = \frac{\partial J_E}{\partial \boldsymbol{Z}_{PE}(t_f)} = -\frac{\boldsymbol{Z}_{PE}(t_f)}{\|\boldsymbol{Z}_{PE}(t_f)\|} \\ \boldsymbol{\lambda}_D(t_f) = \frac{\partial J_D}{\partial \boldsymbol{Z}_{DP}(t_f)} = \frac{\boldsymbol{Z}_{DP}(t_f)}{\|\boldsymbol{Z}_{DP}(t_f)\|} \end{cases} \tag{21}$$

From Equations (20) and (21), we can find the following relationship:

$$
\begin{cases}
\lambda_{PE}(t) = -k_1 \lambda_E(t) \\
\lambda_{DP}(t) = -k_2 \lambda_D(t)
\end{cases}
\tag{22}
$$

In addition, the optimal control strategies need to satisfy:

$$
\begin{cases}
u_D^* = \underset{\|u_D\| \leq 1}{\mathrm{argmin}} H_D \\
u_E^* = \underset{\|u_E\| \leq 1}{\mathrm{argmin}} H_E \\
u_P^* = \underset{\|u_P\| \leq 1}{\mathrm{argmin}} H_P
\end{cases}
\tag{23}
$$

For the sake of brevity, we define new variables $M_i$ $(i = D, E, P)$ as follows:

$$
\begin{cases}
M_D = \lambda_D^T D \Phi T_D \\
M_E = -\lambda_E^T D \Phi T_E \\
M_P = -k_1 \lambda_E^T D \Phi T_P + k_2 \lambda_D^T D \Phi T_P
\end{cases}
\tag{24}
$$

Combining Equations (19), (22), (23), and (24) yields:

$$
\begin{cases}
u_D^* = [u_{xD}^*, u_{yD}^*, u_{zD}^*]^T = -\dfrac{[M_D(4), M_D(5), M_D(6)]^T}{\|[M_D(4), M_D(5), M_D(6)]^T\|} \\[2mm]
u_E^* = [u_{xE}^*, u_{yE}^*, u_{zE}^*]^T = -\dfrac{[M_E(4), M_E(5), M_E(6)]^T}{\|[M_E(4), M_E(5), M_E(6)]^T\|} \\[2mm]
u_P^* = [u_{xP}^*, u_{yP}^*, u_{zP}^*]^T = -\dfrac{[M_P(4), M_P(5), M_P(6)]^T}{\|[M_P(4), M_P(5), M_P(6)]^T\|}
\end{cases}
\tag{25}
$$

Combining Equation (25) and the form of control vector, the optimal control variables are expressed as Equation (26), which satisfies Equation (27).

$$
\begin{cases}
U_D^* = [0, 0, 0, u_{xD}^*, u_{yD}^*, u_{zD}^*]^T \\
U_E^* = [0, 0, 0, u_{xE}^*, u_{yE}^*, u_{zE}^*]^T \\
U_P^* = [0, 0, 0, u_{xP}^*, u_{yP}^*, u_{zP}^*]^T
\end{cases}
\tag{26}
$$

$$
\begin{cases}
J_P(U_P^*, U_E^*, U_D^*) \leq J_P(U_P, U_E^*, U_D^*) \\
J_E(U_P^*, U_E^*) \leq J_E(U_P^*, U_E) \\
J_D(U_P^*, U_D^*) \leq J_D(U_P^*, U_D)
\end{cases}
\tag{27}
$$

Equations (10), (20), (21), and (24)–(26) constitute a TPBVP.

### 3.2. Hybrid Numerical Method

So far, the orbital pursuit-evasion-defense problem has been transformed into a 12-dimensional TPBVP. Generally, this kind of problem cannot be solved analytically, and numerical algorithms must be employed [9]. Numerical algorithms for solving this kind of problems include collocation method [29] and multiple shooting method [30]. The collocation method suffers from poor accuracy and high computational burden, while the multiple shooting method has high accuracy but is very sensitive to the initial guess. As noted by Pontani [9], evolutionary methods constitute an effective statistical search technique for selecting the best parameters. Thus, we apply evolutionary methods to generate the initial guess for the multiple shooting method. A hybrid method combining the multi-objective genetic algorithm and the multiple shooting method is proposed to obtain the solution of the orbital pursuit-evasion-defense game. First, the initial guesses of unknown parameters are obtained by using

the multi-objective genetic algorithm. Then the exact solution of the TPBVP is solved by using the multiple shooting method.

For the sake of clarity, the state equations, the co-state equations, the initial conditions, and the terminal conditions are arranged.

Combining Equations (10), (20), and (26), the state equations and the co-state equations can be concluded as follows:

$$\begin{cases} \dot{\boldsymbol{Z}}_{\mathrm{PE}} = \boldsymbol{D}\boldsymbol{\Phi} T_{\mathrm{P}} \boldsymbol{U}_{\mathrm{P}}^* - \boldsymbol{D}\boldsymbol{\Phi} T_{\mathrm{E}} \boldsymbol{U}_{\mathrm{E}}^* \\ \dot{\boldsymbol{Z}}_{\mathrm{DP}} = \boldsymbol{D}\boldsymbol{\Phi} T_{\mathrm{D}} \boldsymbol{U}_{\mathrm{D}}^* - \boldsymbol{D}\boldsymbol{\Phi} T_{\mathrm{P}} \boldsymbol{U}_{\mathrm{P}}^* \\ \dot{\lambda}_{\mathrm{E}} = 0 \\ \dot{\lambda}_{\mathrm{D}} = 0 \end{cases} \tag{28}$$

The initial conditions of Equation (28) are expressed as follows:

$$\begin{cases} \boldsymbol{Z}_{\mathrm{PE}}(0) = \boldsymbol{D}\boldsymbol{\Phi}(t_{\mathrm{f}}, 0)\boldsymbol{X}_{\mathrm{PE}}(0) \\ \boldsymbol{Z}_{\mathrm{DP}}(0) = \boldsymbol{D}\boldsymbol{\Phi}(t_{\mathrm{f}}, 0)\boldsymbol{X}_{\mathrm{DP}}(0) \end{cases} \tag{29}$$

where $\boldsymbol{X}_{\mathrm{PE}}(0) = \boldsymbol{X}_{\mathrm{P}}(0) - \boldsymbol{X}_{\mathrm{E}}(0)$, $\boldsymbol{X}_{\mathrm{DP}}(0) = \boldsymbol{X}_{\mathrm{D}}(0) - \boldsymbol{X}_{\mathrm{P}}(0)$.

According to Equation (21), the terminal conditions are written as follows:

$$\begin{cases} \lambda_{\mathrm{E}}(t_{\mathrm{f}}) = \frac{\partial J_{\mathrm{E}}}{\partial \boldsymbol{Z}_{\mathrm{PE}}(t_{\mathrm{f}})} = -\frac{\boldsymbol{Z}_{\mathrm{PE}}(t_{\mathrm{f}})}{\|\boldsymbol{Z}_{\mathrm{PE}}(t_{\mathrm{f}})\|} \\ \lambda_{\mathrm{D}}(t_{\mathrm{f}}) = \frac{\partial J_{\mathrm{D}}}{\partial \boldsymbol{Z}_{\mathrm{DP}}(t_{\mathrm{f}})} = \frac{\boldsymbol{Z}_{\mathrm{DP}}(t_{\mathrm{f}})}{\|\boldsymbol{Z}_{\mathrm{DP}}(t_{\mathrm{f}})\|} \end{cases} \tag{30}$$

### 3.2.1. Multi-Objective Genetic Algorithm

In the multi-objective genetic algorithm preprocessing, the terminal time $t_{\mathrm{f}}$ and the unknown initial co-state variables $\lambda_{\mathrm{E}}(0)$ and $\lambda_{\mathrm{D}}(0)$ are taken as parameters (individuals). According to the terminal conditions, the objective functions of the multi-objective genetic algorithm are set as follows:

$$\begin{cases} J_1 = \|\lambda_{\mathrm{E}}(t_{\mathrm{f}}) + \frac{\boldsymbol{Z}_{\mathrm{PE}}(t_{\mathrm{f}})}{\|\boldsymbol{Z}_{\mathrm{PE}}(t_{\mathrm{f}})\|}\| \\ J_2 = \|\lambda_{\mathrm{D}}(t_{\mathrm{f}}) + \frac{\boldsymbol{Z}_{\mathrm{DP}}(t_{\mathrm{f}})}{\|\boldsymbol{Z}_{\mathrm{DP}}(t_{\mathrm{f}})\|}\| \end{cases} \tag{31}$$

The safe distance constraint is applied to ensure that the distance between any two players is greater than the safe distance before the terminal time. The best parameters are obtained by setting the reasonable population size, the appropriate maximum generation, and the suitable operators (i.e., crossover and mutation). The multi-objective genetic algorithm improved by Deb [31] is applied to this problem. This algorithm can reduce the complexity of computation and maintain the diversity of solutions. In this paper, we used the default operators in the toolkit on multi-objective genetic algorithm which is provided by Aravind Seshadri [32]. In addition, the population size and the number of generations are set as 100 and 200 respectively. Because of the use of the multi-objective genetic algorithm, the preprocessing time is relatively long. Thus, this algorithm is suitable for off-line calculation.

### 3.2.2. Multiple Shooting Method

In order to better illustrate the application of the multiple shooting method in this problem, a new state vector is defined:

$$\Omega(t) = [\boldsymbol{Z}_{\mathrm{PE}}(t), \boldsymbol{Z}_{\mathrm{DP}}(t), \lambda_E(t), \lambda_{\mathrm{D}}(t)] \tag{32}$$

Substituting Equation (32) into Equation (28), the system equations can be expressed as follows:

$$\dot{\Omega}(t) = f(t, \Omega(t)) \tag{33}$$

The multiple shooting method transforms the TPBVP into a series of initial value problems. The specific steps are given as follows:

**Step 1.** Divide the time interval $[0, t_f]$ into m subintervals, and $t_k$ $(k = 0, \cdots, m)$ represents the boundary points of subintervals, which satisfy $0 = t_0 < t_1 < \cdots < t_m = t_f$.

**Step 2.** For each subinterval $[t_i, t_{i+1}]$ $(i = 0, \cdots, m-1)$, consider the initial value problem: $\dot{\Omega}(t) = f(t, \Omega(t))$, $\Omega(t_i) = s_i$, where $s_i$ is the initial value of the problem.

**Step 3.** Calculate the initial guess by the multi-objective genetic algorithm.

**Step 4.** Solve the initial value problem on each subinterval to obtain the solution $\Omega(t, t_i, s_i)$.

**Step 5.** Determine whether the condition $\Omega(t_{i+1}, t_i, s_i) = s_{i+1}$ and boundary conditions (29) and (30) are satisfied. If not, use the Newton method to modify the initial value and return to step 4. If the conditions are satisfied, the solution of the TPBVP is obtained successfully.

We point out that the accuracy of the initial guess value affects the solution obtained by the multiple shooting method. If the accuracy of the initial guess value is not enough, the convergence point may not be the desired solution. Moreover, it may increase the number of iterations and prolong the calculation time.

## 4. Results and Discussion

In this section, the following four examples are given to verify the effectiveness of the proposed strategy. Among these, Example 1 and Example 2 are taken as one group. Their initial conditions and maneuver parameters are the same. The differences between the two examples are that when performing orbital maneuvers, the pursuer in Example 1 adopts the control strategy based on the fuzzy comprehensive evaluation, while the pursuer in Example 2 does not consider the impact of the defender, that is, the parameters $k_1 = 1$, $k_2 = 0$ in the objective function $J_P$. Example 3 and Example 4 are taken as the other group, with the differences between the two examples being the same as those between Example 1 and Example 2 in the first group. The initial orbital altitude of their reference orbit $h = 500$ km, the acceleration of gravity $g = 9.8e - 3$ km/s$^2$, and the radius of the Earth $R = 6371.393$ km. During the game, the safety distance between players is set as 0.5 km.

*Example 1.* The maximum unit mass thrusts of the pursuer, the evader, and the defender are $T_P = 0.09 \times g$, $T_E = 0.01 \times g$, and $T_D = 0.02 \times g$, respectively, and the game time is 267.4124 s. The initial positions and velocities of the pursuer, the evader, and the defender are shown in Table 2. The pursuer adopts the control strategy based on the fuzzy comprehensive evaluation.

**Table 2.** Positions and velocities of the initial time.

| Parameter | Pursuer | Evader | Defender |
|---|---|---|---|
| $X$/km | 0 | 12 | 6 |
| $Y$/km | 0 | 16 | 8 |
| $Z$/km | 20 | 0 | 10 |
| $V_X/(\text{km·s}^{-1})$ | 0 | 0 | 0 |
| $V_Y/(\text{km·s}^{-1})$ | 0 | 0 | 0 |
| $V_Z/(\text{km·s}^{-1})$ | 0 | 0 | 0 |

Figure 2 shows the curves of the positions of the three players changing with time in the directions of $X$, $Y$, $Z$. From Figure 2, it can be seen that the pursuer bypasses the interception of the defender and eventually catches up with the evader. From Table 3, it can be seen that at the terminal moment, the distance between the pursuer and the evader is 0.3598 km, which is shorter than the safety distance 0.5 km, indicating that at the terminal moment, the pursuer catches up with the evader. Figure 3 shows the distance between the defender and the pursuer during the game. It reaches the shortest distance at 203.5 s. After that, the distance between the pursuer and the defender becomes longer, the shortest

distance being 0.5099 km, which is longer than the safety distance 0.5 km, indicating that during the game the pursuer successfully bypasses the defender.

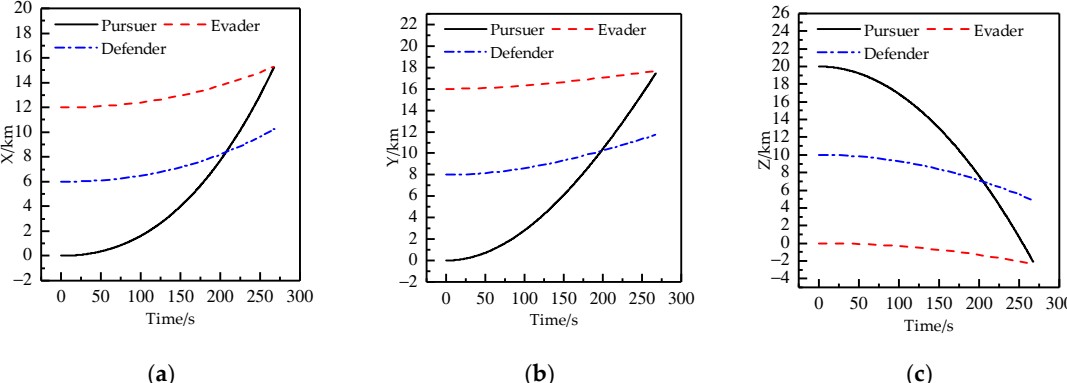

(**a**)　　　　　　　　　(**b**)　　　　　　　　　(**c**)

**Figure 2.** The position of each player changing with time in (**a**) x-axis, (**b**) y-axis, and (**c**) z-axis.

**Table 3.** Position of each player at the end of the game.

| Parameter | Pursuer | Evader | Defender |
|-----------|---------|--------|----------|
| $X$/km | 15.24 | 15.28 | 10.24 |
| $Y$/km | 17.44 | 17.68 | 11.72 |
| $Z$/km | −2.098 | −2.363 | 4.87 |

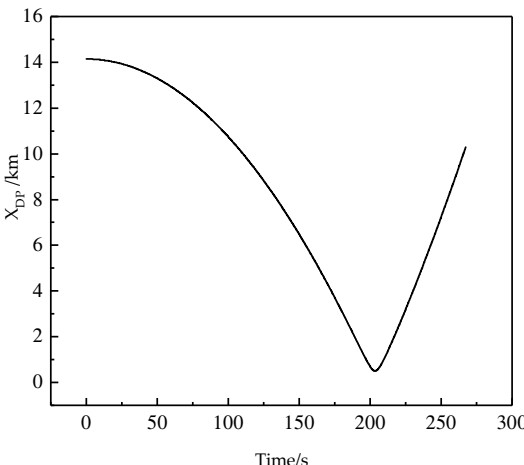

**Figure 3.** The distance between the pursuer and the defender changing with time.

Figure 4 shows the curves of the control variable of each player changing with time in the directions of *X*, *Y*, *Z*. Figure 5 shows the curve of ZEM distance changing with time. From the figures, it can be seen that when the ZEM distance $Z_{DP}(t)$ is close to 0 (i.e., 40 s to 130 s), the pursuer will consider more about evading the defender. So in this phase, the pursuer's control curve is nearer to the control curve of the defender. During the time when the ZEM variables $Z_{DP}(t)$ are not close to 0, the pursuer almost ignores the impact of the defender, so the control curve of the pursuer at this stage almost superposes with that of the evader. Through this strategy, the pursuer successfully bypasses the defenders during the game and finally captures the evader.

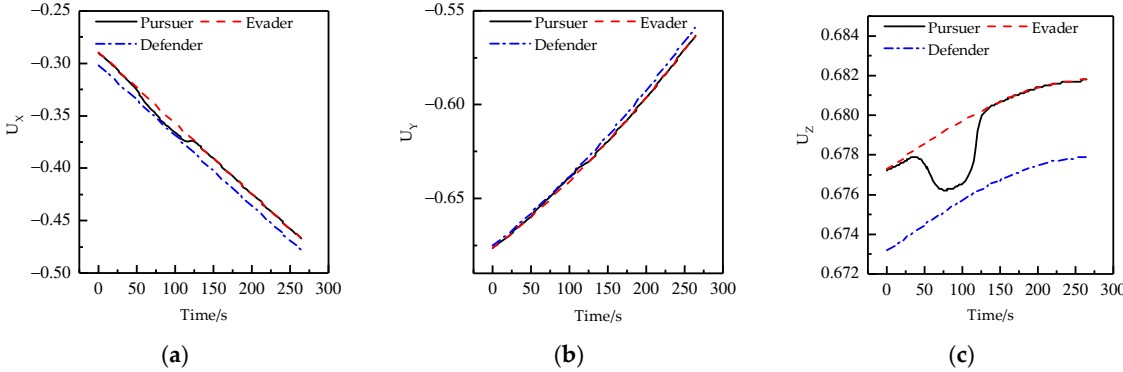

**Figure 4.** The curves of the control variable of each player with time in (**a**) x-axis, (**b**) y-axis, and (**c**) z-axis.

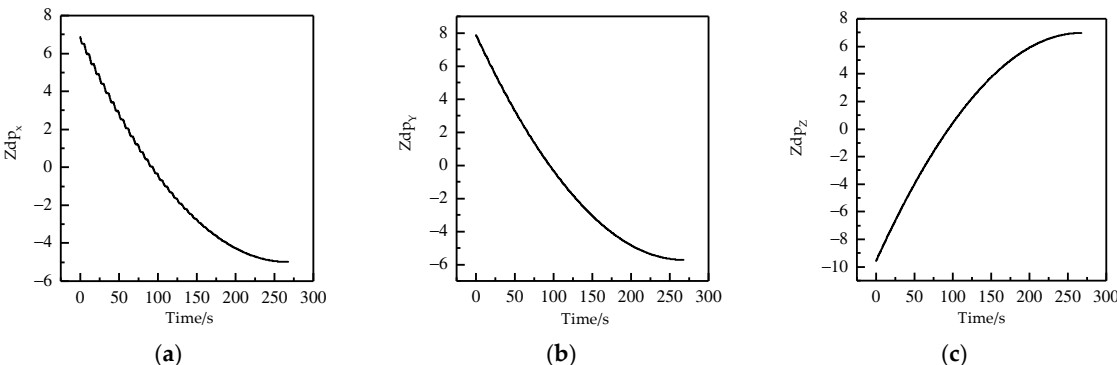

**Figure 5.** The curves of zero-control miss distance with time in (**a**) x-axis, (**b**) y-axis, and (**c**) z-axis.

*Example 2.* The maximum unit mass thrusts of the pursuer, the evader, and the defender are $T_P = 0.09 \times g$, $T_E = 0.01 \times g$, and $T_D = 0.02 \times g$, respectively, and the game time is 159.81193 s. The positions and velocities of the pursuer, the evader and the defender in the initial time are shown in Table 2. The pursuer does not consider the impact of the defender when performing orbital maneuvers.

As shown in Figure 6, the defender successfully intercepts the pursuer at the terminal moment. Figure 7 shows the distance between the defender and the pursuer during the game. According to Figure 7, the distance becomes shorter and shorter in the entire game, which is caused by the pursuer's not considering the impact of the defender. At the terminal moment, the distance between the defender and the pursuer is 0.4004 km, which is shorter than the safety distance 0.5 km.

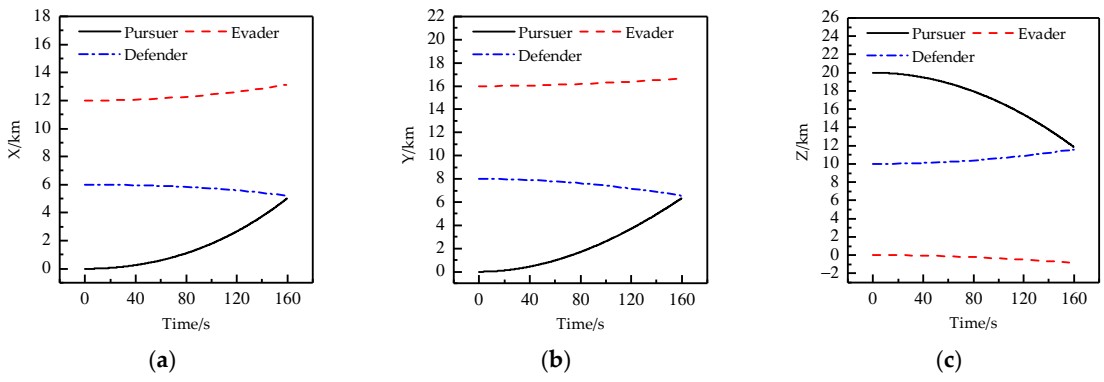

**Figure 6.** The position of each player changing over time in (**a**) x-axis, (**b**) y-axis, and (**c**) z-axis.

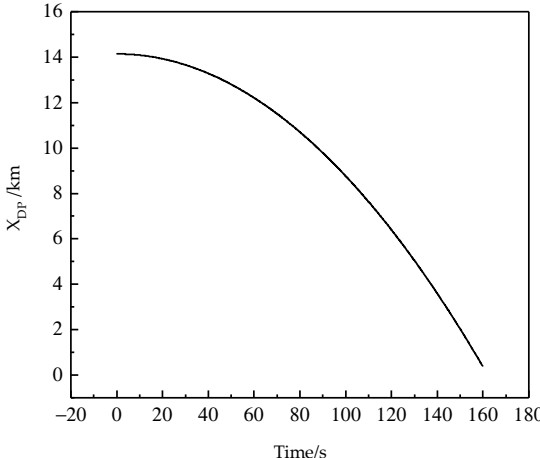

**Figure 7.** The distance between the pursuer and the defender changing over time.

Figure 8 shows the control variable of each player changing with time in the game. As shown in the figure, the control curve of the pursuer overlaps with that of the evader in the whole procedure, the reason being that the pursuer only considers the evader when performing orbital maneuvers.

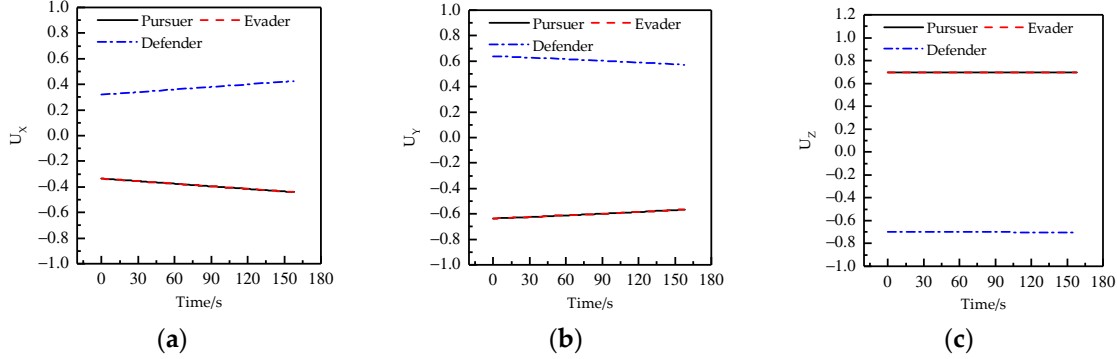

**Figure 8.** The control variable of each player changing over time in (**a**) x-axis, (**b**) y-axis, and (**c**) z-axis.

Comparing Example 1 with Example 2, it can be seen that with the control strategy based on the fuzzy comprehensive evaluation, the pursuer can successfully bypass the defender, and finally capture the evader. The pursuer, for not considering the impact of the defender, is eventually intercepted by the defender.

***Example 3.*** The maximum unit mass thrusts of the pursuer, the evader, and the defender are $T_P = 0.14 \times g$, $T_E = 0.01 \times g$, and $T_D = 0.14 \times g$, respectively. The maneuverability of the defender and that of the pursuer are the same, and the game time is 177.87788 s. The positions and velocities of the pursuer, the evader, and the defender in the initial time are shown in Table 4. The pursuer adopts the control strategy based on the fuzzy comprehensive evaluation.

**Table 4.** Positions and velocities of the initial time.

| Parameter | Pursuer | Evader | Defender |
|:---:|:---:|:---:|:---:|
| $X/\text{km}$ | 0 | 8 | 18 |
| $Y/\text{km}$ | 0 | 9 | 24 |
| $Z/\text{km}$ | 30 | 12 | 0 |
| $V_X/(\text{km·s}^{-1})$ | 0 | 0 | 0 |
| $V_Y/(\text{km·s}^{-1})$ | 0 | 0 | 0 |
| $V_Z/(\text{km·s}^{-1})$ | 0 | 0 | 0 |

Figure 9 shows the curves of the positions of the three players changing with time in directions of *X*, *Y*, *Z*. As shown in Figure 9, the defender successfully intercepts the pursuer at the end of the game. Figure 10 shows the curves of the control variable of each player changing with time in the directions of *X*, *Y*, *Z*. At 160 s or so, the pursuer starts to change the control strategy to evade the defender. However, because of the same maneuverability of the defender and the pursuer, the pursuer does not successfully bypass the interception of the defender. Table 5 shows the position of each player at the terminal moment. From Table 5, it can be seen that at the terminal moment, the distance between the defender and the pursuer is 0.4223 km, which is shorter than the safety distance 0.5 km, and at the terminal moment, the distance between the pursuer and the evader is 1.7823 km, which is longer than the safety distance 0.5 km. All the above show that at the terminal moment, the defender successfully intercepts the pursuer and that the evader successfully evades the capture of the pursuer.

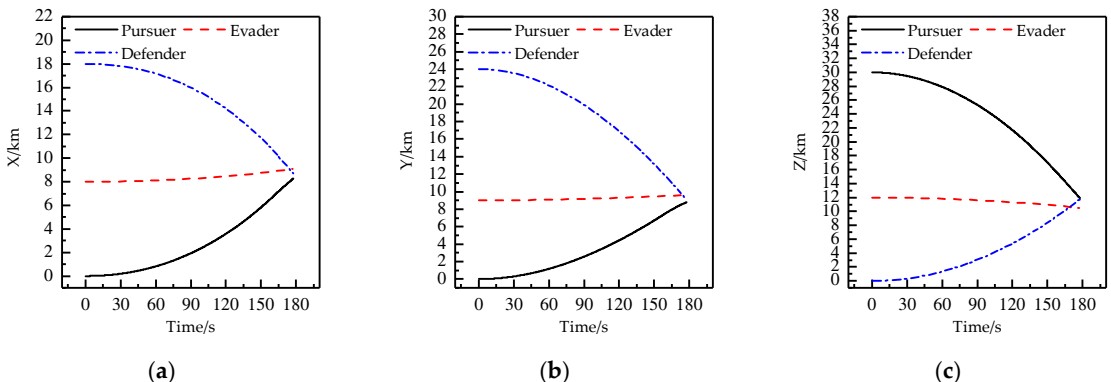

**Figure 9.** The position of each player changing over time in (**a**) x-axis, (**b**) y-axis, and (**c**) z-axis.

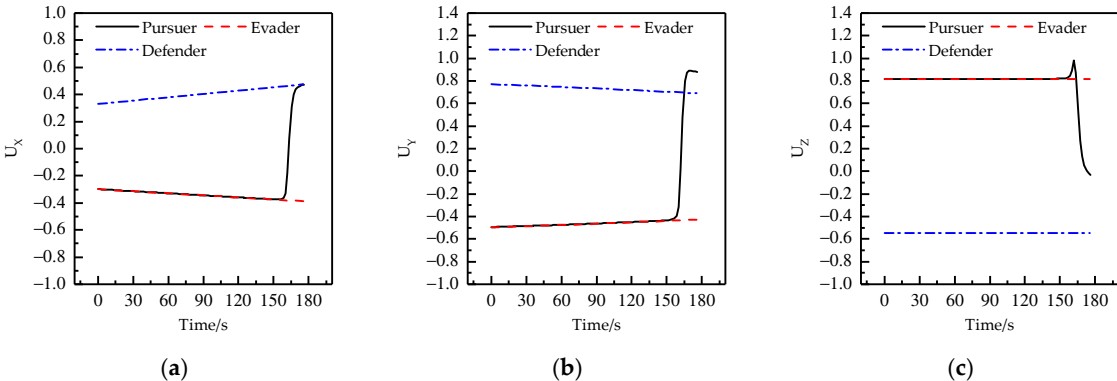

**Figure 10.** The control variable of each player changing over time in (**a**) x-axis, (**b**) y-axis, and (**c**) z-axis.

**Table 5.** Position of each player at the end of the game.

| Parameter | Pursuer | Evader | Defender |
|-----------|---------|--------|----------|
| *X*/km | 8.293 | 9.083 | 8.638 |
| *Y*/km | 8.804 | 9.609 | 9.026 |
| *Z*/km | 11.89 | 10.51 | 11.79 |

***Example 4.*** The maximum unit mass thrusts of the pursuer, the evader, and the defender are $T_P = 0.14 \times g$, $T_E = 0.01 \times g$, $T_D = 0.14 \times g$, respectively. The maneuverability of the defender and that of the pursuer are the same, and the game time is 177.19431 s. The positions and velocities of the pursuer, the evader, and defender in the initial time are shown in Table 4. The pursuer does not consider the impact of the defender when performing orbital maneuvers.

As shown in Figure 11, the defender intercepts the pursuer at the terminal moment. From Table 6, it can be seen that the distance between the pursuer and the defender in the game is 0.3959 km, which is shorter than the safety distance 0.5 km, and that the distance between the pursuer and the evader is 1.6016 km, which is longer than the safety distance 0.5 km. This shows that the defender intercepts the pursuer successfully at the terminal moment, and the evader evades the capture of the pursuer successfully. Figure 12 shows the curves of the control variable of each player changing with time in the directions of *X*, *Y*, *Z*. From the figure, it can be seen that the control curves of the pursuer remain overlapped with those of the evader.

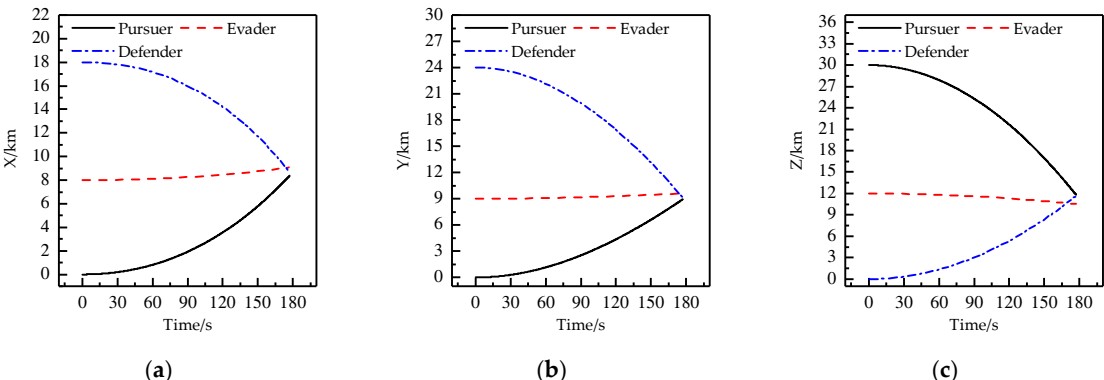

**Figure 11.** The position of each player changing with time in (**a**) x-axis, (**b**) y-axis, and (**c**) z-axis.

**Table 6.** Position of each player at the end of the game.

| Parameter | Pursuer | Evader | Defender |
|---|---|---|---|
| $X$/km | 8.384 | 9.069 | 8.714 |
| $Y$/km | 8.956 | 9.593 | 9.132 |
| $Z$/km | 11.81 | 10.51 | 11.68 |

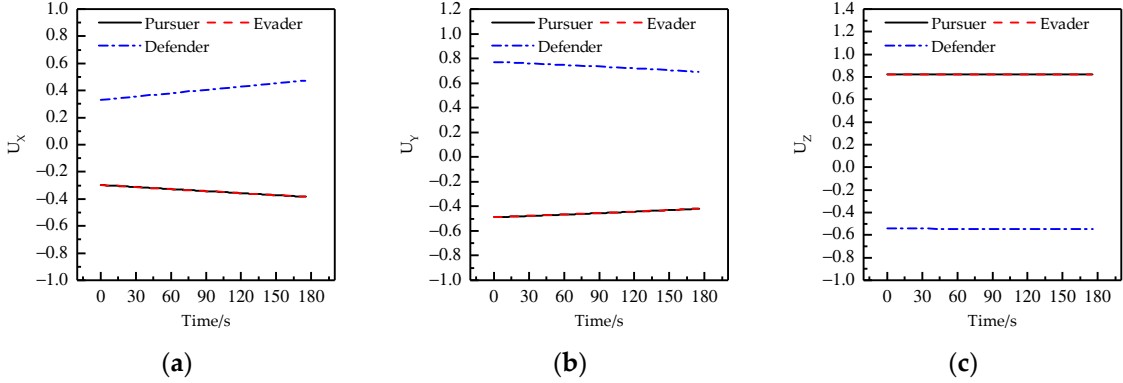

**Figure 12.** The control variable of each player changing over time in (**a**) x-axis, (**b**) y-axis, and (**c**) z-axis.

Comparing Example 3 with Example 4, it can be seen that, because of the different control strategies adopted by the pursuer, the time that the defender takes to intercept the pursuer in Example 3 is longer than that in Example 4. Moreover, at the terminal moment, the distance between the pursuer and the defender in Example 4 is shorter than that in Example 3.

The comparison between Example 1 and Example 2 shows that when the control variable of the pursuer is in a dominant position, the optimal control strategy proposed in this paper makes the pursuer bypass the defender and capture the evader. The comparison between Example 3 and Example 4 shows that when the control variable of the pursuer is not in a dominant position, the optimal control strategy proposed in this paper prolongs the time that the defender takes to intercept the pursuer.

## 5. Conclusions

The fuzzy comprehensive evaluation and the differential game theory are applied to design the control strategy of the pursuer in the orbital pursuit-evasion-defense problem. The hybrid method combining the multi-objective genetic algorithm and the multiple shooting method is proposed to solve the problem. The simulation results show that when the pursuer control is in a dominant position, the control strategy proposed in this paper can make the pursuer bypass the defender and capture the evader, and that when the pursuer control is not in a dominant position, the control strategy proposed in this paper can prolong the time that the defender takes to intercept the pursuer. The proposed control strategy is applicable to the orbital pursuit-evasion-defense scenario, in which the players adopt the continuous low thrust propulsion. When the ZEM distance between the pursuer and the defender is close to zero, the control strategy can be automatically switched to parallel with the defender's control strategy, so that the pursuer can effectively avoid the interception of the defender.

However, the limitation of this paper is that the terminal time of the game is given by the genetic algorithm, which is not accurate. Further research will be carried out on the accurate calculation of the terminal time.

**Author Contributions:** J.Z. and L.Z. conceived the framework and structured the paper; J.Z. and J.C. performed the experiments and analyzed the data; J.Z., S.W., and Y.W. wrote and revised the paper.

**Funding:** This research was jointly funded by the National Natural Science Foundation of China (Nos. 61633008, 61773132, 61803115), the 7th Generation Ultra Deep Water Drilling Unit Innovation Project sponsored by Chinese Ministry of Industry and Information Technology, the Heilongjiang Province Science Fund for Distinguished Young Scholars (No. JC2018019), and the Fundamental Research Funds for Central Universities (No. HEUCFP201768).

**Acknowledgments:** We gratefully acknowledge Aravind Seshadri for providing the toolkit on multi-objective genetic algorithm at the following website: https://www.mathworks.com/matlabcentral/fileexchange/10429-nsga-ii-a-multi-objective-optimization-algorithm.

**Conflicts of Interest:** The authors declare no conflict of interest.

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
