# Peer review of "Pursuer’s Control Strategy for Orbital Pursuit-Evasion-Defense Game with Continuous Low Thrust Propulsion"

_applsci, doi:10.3390/app9153190_

Round 1

Reviewer 1 Report

You're written a very good manuscript, very well written describing very interesting research.

I did notice some minor typographical errors that auto-correct features in Microsoft Word would miss, e.g. in line 130, in stating "though", I think you meant to say "although".  No big deal.  I'm sure this paper will be published, so I recommend you take some time to one-last time read the paper word-by-word very slowly, so the final published paper reflects good upon you and upon Applied Sciences.

Regarding content, I have only one recommendation:  While readers will be sufficiently familiar with using relative coordinates, since they prevail in many engineering disciplines, I don't think the readers will understand zero effort miss (ZEM) variables even after reading your paper.  I recommended adding some foreshadowing explanatory sentences in section 1 with some citations as well. Consider expanding the explanation in section 2.1 as well.  Thank you.

I'm happy to share with you my "private comments" to the Editor in the next paragraph. 

__________________________________________________________________________

This is a good paper, and Applied Sciences should be proud to publish it.  It is well written (with some minor typographic errors) and very well articulated, e.g. the step-by-step methods elaborated in section 3 are key to the paper's merit by aiding duplicability.  The use of tabularized results is also a strength. Some of the (very good) plots have borderline-too small font.  I will urge the authors to use the figure captions' font size as a minimum to aid readability.  I have not given too many "accept in present form" recommendations, but I do so on this paper with the faith the authors will make minor modifications during typesetting. 

Reviewer 2 Report

A good theoretical paper with some interesting results. Worthy of publication to disseminate the ideas.

1) Will the authors please comment on how this algorithm could be implemented in a robust way in real time? This will be an issue if you use GA.

2) You run the Multi-Objective GA (MOGA) for how many iterations before settling on the initial guess of parameters?

3) How does the solution depend on the accuracy of the initial guess? Should have a qualitative discussion, at least. How sensitive is it to parameter variation?

4) What is the "reasonable population size", mutation parameter, and style of GA? The description is too brief. I ask this because speed of convergence to a solution will beimportant in this type of problem, and not only that a solution can be found.

Round 2

Reviewer 2 Report

Paper is ok as a contribution. Authors have addressed my concerns.

Please provide the link to the GA toolkit in your acknowledgements. This will enable other researchers to evaluate your results and further the state of art.

Please also list the population and that you used the default settings in the toolkit. And just add your answers to my queries regarding the GA in the paper, as short sentences regarding the GA used.
